# Experimental and Numerical Mechanical Characterization of Unreinforced and Reinforced Masonry Elements with Weak Air Lime Mortar Joints

Giuseppe Brando * , Gianluca Vacca, Francesco Di Michele, Ilaria Capasso and Enrico Spacone

Department of Engineering and Geology, University "G. d'Annunzio" of Chieti-Pescara, 65127 Pescara, Italy; gianluca.vacca93@gmail.com (G.V.); dimichelefrancesco@live.it (F.D.M.); ilaria.capasso@unich.it (I.C.); espacone@unich.it (E.S.)
* Correspondence: giuseppe.brando@unich.it; Tel.: +39-0854537253

**Abstract:** This paper deals with the results of an experimental and numerical campaign aimed at characterizing the mechanical response of masonry components and panels made of limestone units kept together by weak air lime mortar joints. The selected air lime mortar, typical of ancient masonry buildings but difficult to be built-up artificially, was specifically prepared for the experimental analyses, with the aim of obtaining a laboratory compression strength of 0.25–0.50 MPa. In the first part of the paper, the performed tests concerning the strength of the units (mean compression strength of 80 MPa) and of the mortar (mean compression strength after 28 days of 0.30 MPa), are described for different curing periods. Moreover, tests of masonry triplets in shear (shear strength of 0.11 MPa for null axial forces) are shown and used in order to establish the main parameters of the Mohr–Coulomb failure criterium. Then, the calibration of a continuous numerical micro-model implemented in Kratos Multiphysics is presented. The model is used for reproducing the behavior of an unreinforced panel in shear made of the studied masonry and to appraise the effectiveness of a FRCM- (Fiber Reinforced Cementitious Matrix) based reinforcement intervention applied. The obtained results proved that FRCM allows to increase the strength of the considered masonry type by about eight times and the ductility by about thirteen times.

**Keywords:** air lime mortar; masonry; micro-models; FRCM; masonry retrofitting; sustainable retrofitting



## 1. Introduction

Many people in seismic-prone areas live in old historic centers made of low-rise buildings (mostly 2/3 stories) characterized by masonries with poor quality mortar [1,2]. Enhancing the seismic response of these buildings is of paramount importance, both for public safety and for increasing the tourist attractivity of the historic centers that these buildings populate.

The structural behavior of a masonry wall depends on several parameters [3]. However, an exhaustive literature analysis proved that mortar joints definitely play the most influencing role in determining, under lateral loads, both the arising of the first cracks and the collapse response [4–6], particularly in the presence of mortar characterized by low strength [7,8].

Kariotis et al. [9] carried out one of the first experimental analyses on real brick walls with low-strength mortar belonging to an unreinforced masonry building that had to be demolished. The obtained results allowed to give recommendations, in particular with respect to the tensile bond strength to be assumed at the mortar–unit interface. With reference to experimental tests on real buildings, more recently, Costa et al. [10] presented a test campaign on five masonry panels belonging to a building stricken by the 1998 Azores earthquake. The tested walls were made of a double leaf basalt stone masonry with poor infill and weak mortar joints. The experiments were finalized to characterize both the

in-plane and the out-of-plane behavior of the walls, as well as to judge the effectiveness of some retrofitting techniques. On the same line, Fonti et al. [11] presented the results of three different campaigns carried out on weak mortar masonry walls of three existing buildings, with the purpose of developing theoretical models able to interpret the out-of-plane behavior of a wall under cyclic forces. Other experimental campaigns on real masonry elements made of low-strength mortar were carried out by Corradi et al. [12], who performed compression tests, diagonal compression tests and shear-compression tests on masonry panels extracted from several buildings hit by the 1997/1998 seismic sequence which occurred in Umbria and Marche (Italy). Andreini et al. [13], Uranjiek et al. [14] and many other researchers recognized the importance of better investigating the response of masonry elements when mortar joints are the weakest component far.

Although in situ tests on as-built structures are definitely the best way to obtain realistic information on the experimented masonry, often they are unpractical, in particular when they concern ancient buildings that, belonging to historical centers, cannot be subjected to destructive tests [15,16]. Laboratory tests and validated numerical models are therefore necessary, but, in this case, reproducing the same properties of a deteriorated carbonated mortar, at least from a mechanical point of view, is the most challenging task. Many authors worked on this issue. Kepler [17] developed mixed designs of low-strength mortar to be used as structural backfill, including compressive strength, flowability and set time. Drougkas et al. [18] performed an experimental campaign for assessing the response of brick masonry with lime mortar joints. Two types of mortar were considered for the purpose, both typical of the vast majority of historical and existing masonry structures: air lime mortar and hydraulic lime mortar combined with fine sand and without Portland cement content. The composition of the two mortars led to a compressive strength, measured after 49 days, of about 1.2 Mpa for the air lime mortar and of 1.8 MPa for the hydraulic air mortar. Angiolilli and Gregori [19] carried out triplet tests, working on specimens made of irregular calcareous stone units, gathered among the debris of some collapsed buildings after the 2009 L'Aquila earthquake. They prepared a special mortar, after several attempts, mixing commercial hydraulic lime mortar, local crushed limestone sand and local natural clay in a composition of 1:2:1. This mortar was characterized by a compressive strength of almost 2.0 MPa. A Chinese study, carried out by Huang et al. [20], reported an experimental campaign focused on the compressive strength of masonry coupons made of concrete units tied with low-strength lime mortar, showing a brittle behavior and suggesting that, for this type of masonry, it is of fundamental importance to keep the compressive stress under a critical value to be selected according to the mortar strength.

From a numerical point of view, in [21] Sarhosis and Sheng calibrated a distinct element model for low bond strength masonry, developing the structural virtual twins of some experimental coupons by minimizing the differences, in terms of shear behavior, between the numerical and the test results, via optimization processes. On the basis of this model, the same Authors [22] investigated, through a parametric study, the effect of the mortar strength on the response of a wall, proving that the tensile strength is the most influencing factor affecting the arising of the first cracks, whereas cohesive strength and friction angle determine the behavior of the panel from the onset of cracking up to collapse.

In the research field described above, this paper proposes an experimental and numerical study concerning the structural response of reinforced and FRCM reinforced masonry elements made of limestone units and low-strength air lime mortar, which are typical of many old buildings of a large plethora of historical centers [23,24] in the Mediterranean area.

First, in Section 2.2, the experimental tests carried out on the used limestone units, on the mortar and on triplets in shear are presented. In the same section, the composition used for the low-strength mortar, as well as the preparation phases, are discussed. In fact, as highlighted previously, this is one of the most relevant issues to be faced in the laboratory. On the basis of the experimental results, a numerical model implemented in Kratos Multiphysics [25], is presented in Section 2.3. Its reliability is proven by reproducing

the tests on triplets in shear. Then, the model is used in order to simulate diagonal shear tests on unreinforced and FRCM reinforced square panels made of the considered masonry type.

To the best of our knowledge, the paper deepens a type of masonry, with very weak mortar joints, that, although certain literature on the topic exists, was seldom investigated, particularly in the laboratory environment, where obtaining very a low strength of the mortar (about 0.5 MPa) has often represented a challenging objective. Furthermore, the evaluation of the effectiveness of FRCM-based retrofitting intervention on this type of masonry definitely represents an element of significant relevance in the considered research field, as well as an important advancement with respect to the current literature.

## 2. The Experimental Tests

### 2.1. General

Masonry materials and construction techniques considered for the study presented herein have been selected considering a typical Italian area of the Apennines, in order to recreate conditions that are representative of what can actually be found in existing buildings.

The experimental campaign consisted of:

- Compression tests on regular limestone blocks (calcareous natural stones);
- Compression and flexural tests on a special air lime mortar conceived to reproduce the same mechanical features of real buildings;
- Direct Shear tests on masonry triplets made of the two materials listed above.

In the following, the outcomes of the tests are shown and discussed.

### 2.2. Compression Tests on Stone Units

The selected units are regular limestone blocks (calcareous natural stones) typical of Pacentro, in the district of L'Aquila, in Italy. This stone was widely used in the past for several typologies of buildings, such as churches (Figure 1a) and dwellings (Figure 1b), because of their intrinsic resistance to freezing cycles.

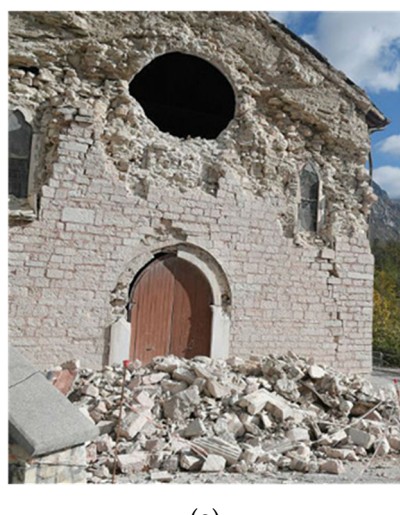
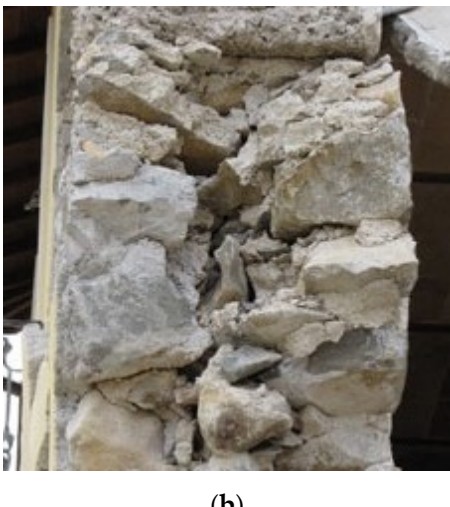

(**a**)　　　　　　　　　　　　　　　　　　　(**b**)

**Figure 1.** Masonry constructions in Italy made of Pacentro stones: (**a**) the Santa Maria Assunta church observed in Ussita after the 2016 Central Italy Earthquake and (**b**) damaged masonry buildings observed in L'Aquila after the 2009 Earthquake.

Ten $70 \times 70 \times 70$ [mm] cube stone units, characterized by a measured mass density of $\rho = 2458$ kg/m$^3$, were subjected to compression tests, according to the European Standard EN 1926:2000 [26]. The obtained compressive strengths are reported in Figure 2. They gave back a mean value of $fc = 79.71$ MPa and a standard deviation of $\delta = 4.5$ MPa. Moreover, the performed tests allowed to measure a mean elastic normal modulus of $E = 20,200$ MPa.

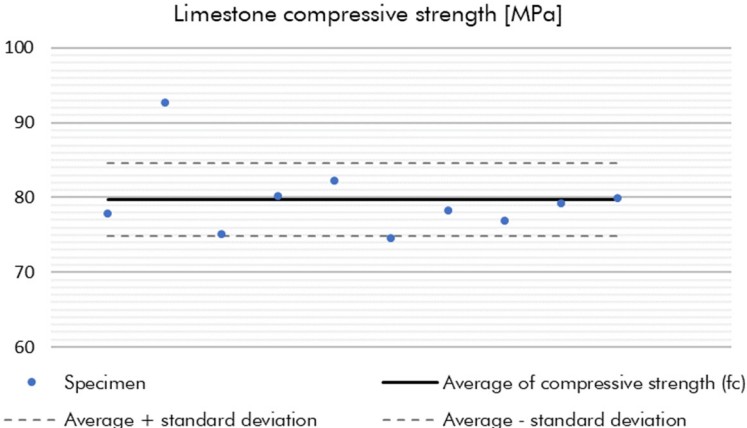

**Figure 2.** Compressive strength (average plus or minus standard deviation) of the stone material.

### 2.3. Tests on Mortar

As stated previously, one of the main issues in laboratory tests of ancient masonry is the reproduction of a mortar characterized by degraded mechanical features.

To this purpose, a special mix was prepared according to the composition in Table 1, targeting a maximum compressive strength ranging between 0.25 MPa and 0.50 MPa. The obtained air mortar is mostly composed of lime derived from crushed limestone and natural aggregates with granulometry between 4 mm and 16 mm. The mix was completed with a superplasticizer (Rheobuild 1–2), in order to achieve the desired low strength.

**Table 1.** Composition of mortar that used in the present work.

| Components | % in Weight | Weight for 20 kg |
|---|---|---|
| Powdered lime | 31.15 | 6.23 kg |
| Natural aggregates (4 ÷ 16 mm) | 68.54 | 13.71 kg |
| Rheobuild 1–2 (© BASF) | 0.31 | 0.06 kg |
| Water | | 5 kg |
| SUM | 100 | 20 + 5 kg of water |

XRD and TGA/DTGA analyses were carried out in order to identify and characterize the hardening products of all the experimental mortars produced. In particular, XRD analyses were performed on powdered samples using a Panalytical X'Pert Pro diffractometer equipped with PixCel 1D detector (operative conditions: CuKα1/Kα2 radiation, 40 kV, 40 mA, 2Q range from 5 to 80° step size 0.0131° 2Φ, counting time 40 s per step) and simultaneous thermal analyses using a Netzsch STA409PC Luxx apparatus (temperature range: 20–1000 °C; heating rate: 10 °C/min; atmosphere: $N_2$). The experimental results are reported in Figure 3.

From the XRD spectra (Figure 3a) it is possible to infer that the main mineralogical phase present in the mortar samples, for all curing times, is calcite (calcium carbonate, $CaCO_3$) which can be considered as clear evidence that the air lime hardening process has occurred. Furthermore, in the XRD spectra, there are also peaks identified as unreacted lime, revealing the presence of hydrated lime ($Ca(OH)_2$) which did not take place in the carbonation process. The thermograms of mortar samples at 7, 14 and 28 days (Figure 3b) showed one main weight loss, located at around 800 °C and equal to 36.22%, 36.79% and 38.76%, respectively.

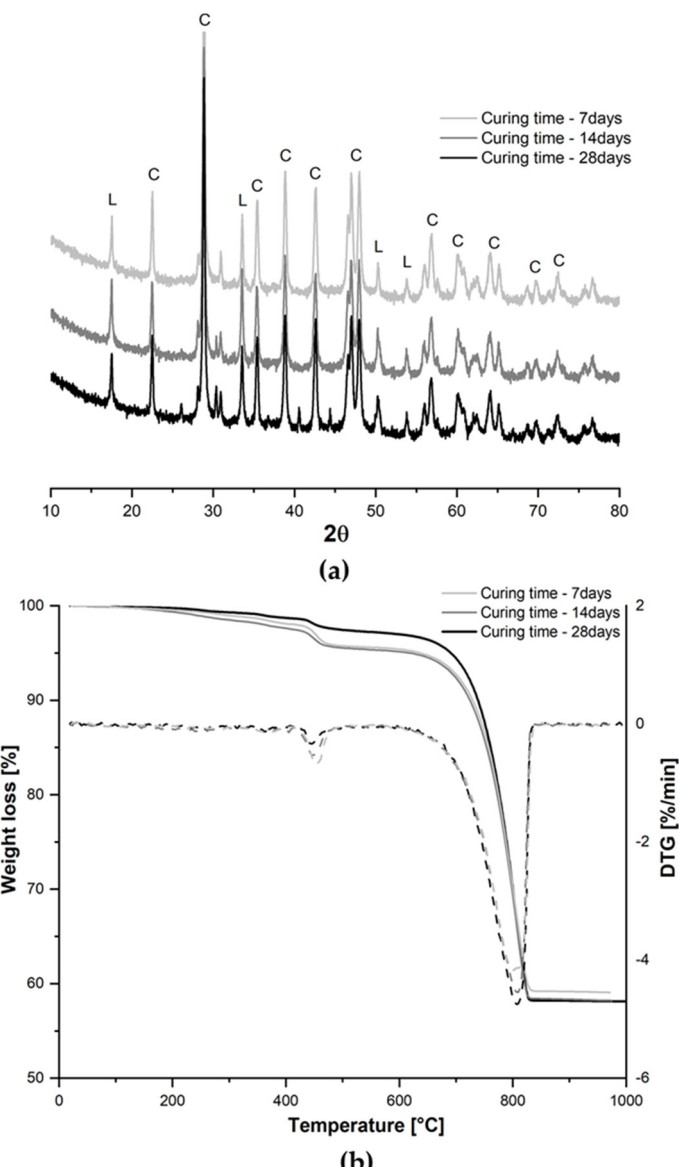

**Figure 3.** (**a**) XRD spectra (C = Calcite, L = unreacted lime) and (**b**) thermogravimetric analyses (straight lines) with their first derivative (dotted lines) of the mortar at 7, 14 and 28 days.

This weight loss is due to the decomposition of calcite ($CaCO_3$) and confirms that the carbonation process of air lime successfully happened, leading to the hardening of mortar samples. It is worth noting that the carbonation process proceeds until reaching 28 days, as clearly proved by the growing values of weight losses due to calcite for mortar with growing curing times. Moreover, inspecting the thermograms, it is possible to notice at about 450 °C, the presence of small weight losses of 2.35%, 2.09% and 1.28% for mortar samples at 7, 14 and 28 days, respectively, which can be related to the de-hydration process of ($Ca(OH)_2$), further confirming the presence of traces of unreacted lime, already visible from the XRD results.

The main mechanical characteristics of the mortar have been obtained by means of standardized laboratory compression and flexural tests, carried out according to the European Standard EN 1015-11:2019 [27]. Furthermore, the tests were repeated at different curing times of 7, 14 and 28 days. In Figure 4, the manufactured specimens are shown.

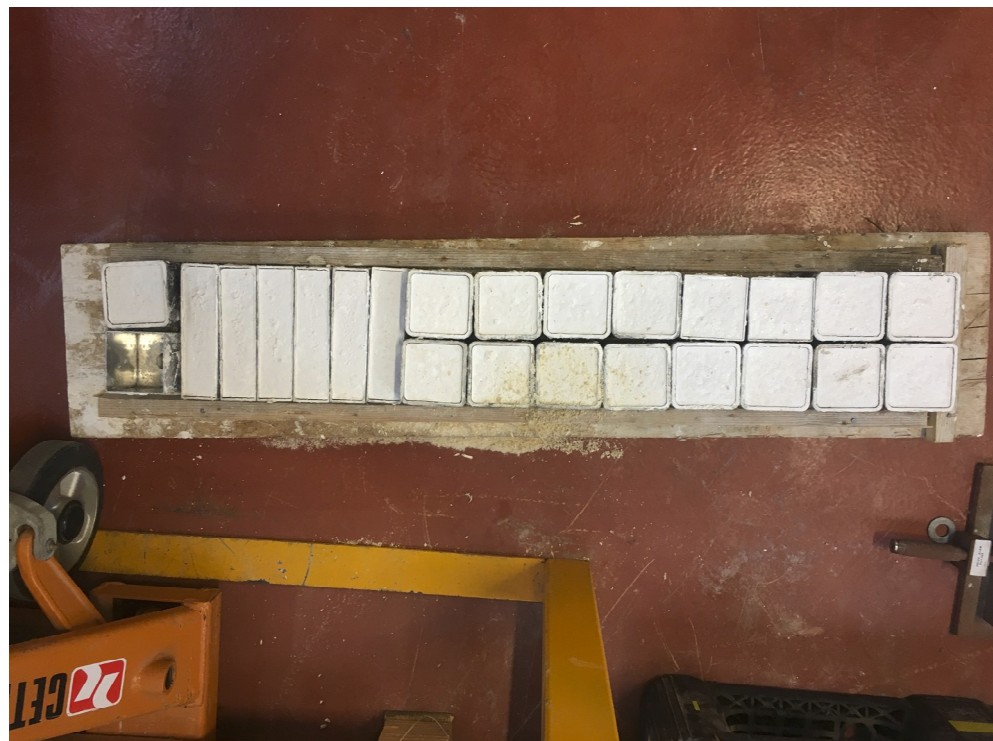

**Figure 4.** The mortar specimens manufactured for compression and bending tests.

As for the compression tests, seven mortar cubes ($70 \times 70 \times 70$ [mm]) for curing times of 7 and 28 days have been considered. Instead, three mortar cubes only for 14 days curing times have been tested: in fact, four coupons have been excluded from the experimental campaign because they were compromised due to spalling when they were extracted from the formworks.

The obtained results are reported in Figure 5a, where the crushed cubes at the end of some tests are also depicted. In Figure 5b, the mean value of the strengths, together with the related standard deviations are shown. As it is possible to observe, a mean strength of 0.19 MPa (standard deviation of 0.03 MPa), 0.22 MPa (standard deviation of 0.005 MPa) and 0.29 MPa (standard deviation of 0.01 MPa) were obtained at 7, 14 and 28 days, respectively, with a curing process following a linear trend. This means that the selected mix complied with the design objective of having a strength in the range of 0.25–0.50 MPa.

As for the flexural tests, three point-loading have been performed on three coupons for each curing time. These gave back (Figure 6) a tension strength $f_t$ of 0.058 MPa (standard deviation of 0.03), 0.067 MPa (standard deviation of 0.01) and 0.082 MPa (standard deviation of 0.03), at 7, 14 and 28 days, respectively, with a curing process following a parabolic trend.

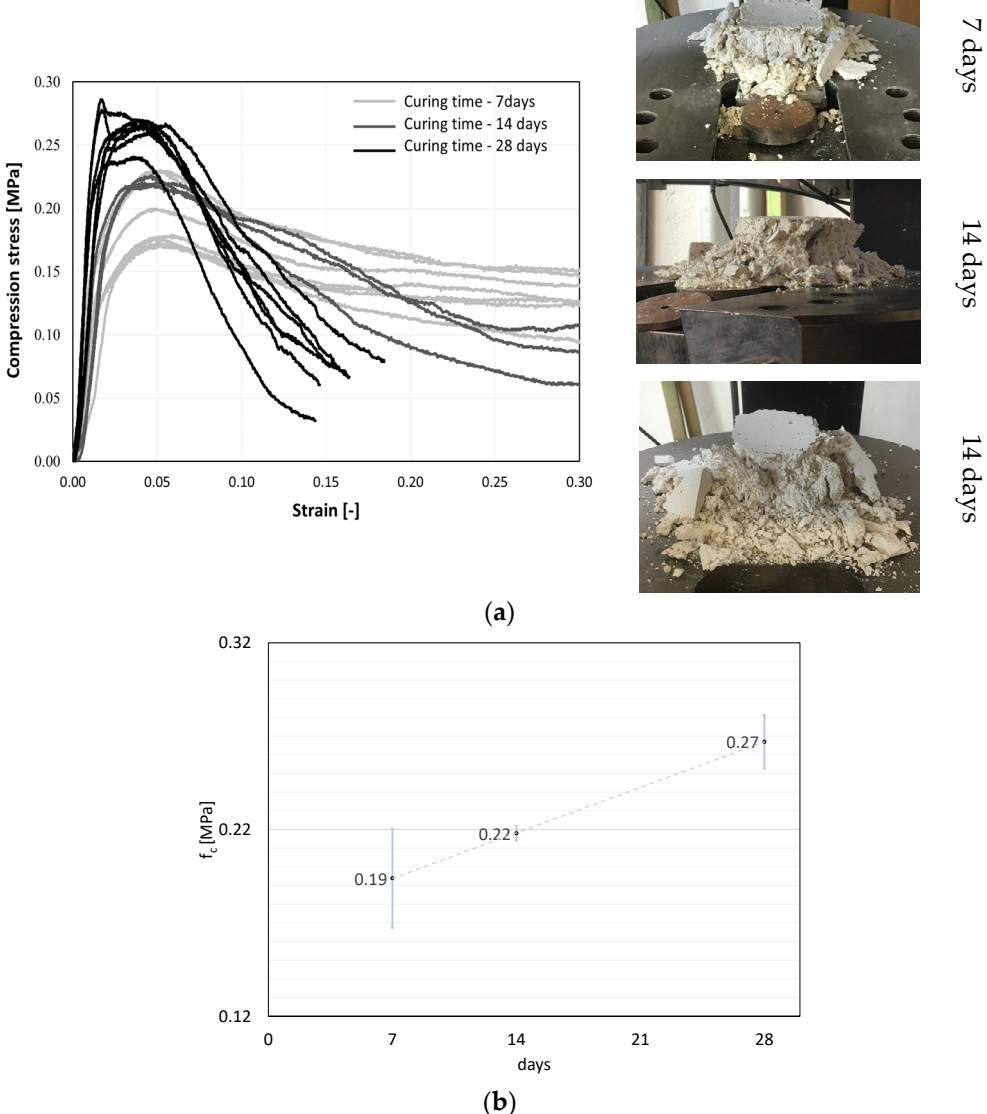

(**a**)

**Figure 5.** (**a**) Stress–strain diagrams and (**b**) compressive strengths (mean values plus or minus a standard deviation) of the mortar at 7, 14 and 28 days.

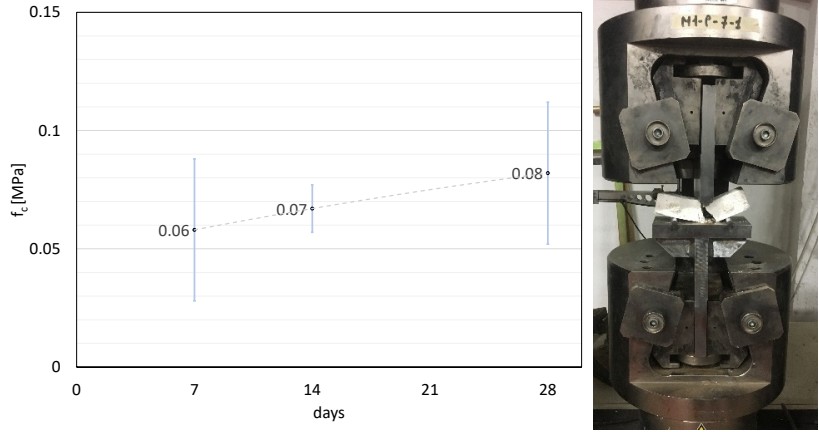

**Figure 6.** The bending strength of the mortar through three-point-bending tests.

### 2.4. Direct Shear Stress on Triplets

In order to evaluate the bond shear strength at the unit–mortar interfaces, seven masonry triplets were built and tested under shear, according to EN 1052-3:2002 and EN 1052-3:2007 [28,29]. The sizes of the specimens were $350 \times 158 \times 191$ mm, formed by three $110 \times 158 \times 191$ stone units and two mortar joints characterized by a thickness of 10 mm.

The triplets were laid on a rigid steel base, which was shaped so to be in contact only with the external stones, whereas a gap between the steel base itself and the central stone was assured (Figure 7a). This gap was necessary to leave the central stone free to move laterally during the loading process. In order to apply pre-selected levels of compression (0.22 MPa, 0.38 MPa and 0.60 MPa), axial forces on the triplets were applied through two stiffened steel plates connected by pre-tensioned bars, in contact with the outer stones of the triplets. The internal forces on the bars were applied, through a torque wrench, by increasing of 0.5 kN the axial force of each bar (corresponding to a total increment of stress of 0.06 MPa for each round of tightening).

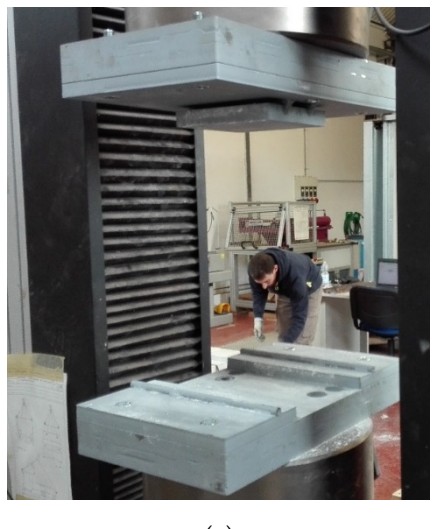
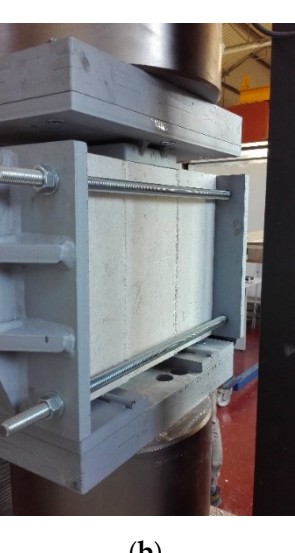

(**a**)　　　　　　　　　　　　　　　　　　(**b**)

**Figure 7.** Set-up of direct shear test on triplet: (**a**) The rigid base on which the triplets laid; (**b**) the post-tensioned bars for applying different levels of compression.

The universal machine Galbabini "Sun 60" was used to apply lateral displacements, with a velocity of 0.01 mm/s, on the central stone (Figure 7b), so to produce shear on the two mortar joints.

In Figure 8, the sliding failures observed for three of the tested specimens for different compression stresses, are shown. Obviously, the stone units were not involved in the failure and a perfect sliding in the mortar joint was observed.

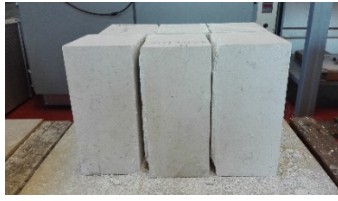
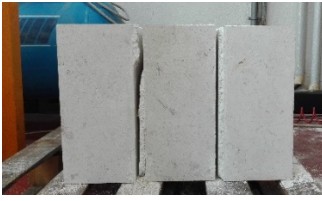
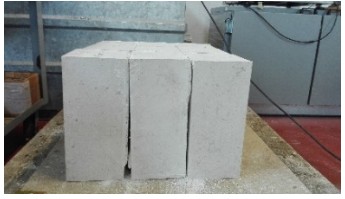

*Normal stress: 0.22 MPa*　　　*Normal stress: 0.38 MPa*　　　*Normal stress: 0.60 MPa*

**Figure 8.** The observed failure modes of triplets for different axial preloads.

Table 2 reports the failure shear stress $\tau_u$ obtained for different compression stress $\sigma_0$. The experimental results have been plotted in Figure 9, together with the mean values. As it is possible to observe, these mean values can be well interpolated by a linear law

obtained by applying a linear regression model with a coefficient of determination $R^2$ equal to 0.97 (0.84 if taken with respect to all the experimental measures). The obtained linear law allows us to interpret the obtained results in terms of Mohr–Coulomb failure criterium, with a cohesion of 0.11 MPa, and a friction coefficient of 0.22.

**Table 2.** Results obtained by direct shear test.

| Specimens | $\tau u$ [MPa] | $\sigma_0$ [MPa] |
|:---:|:---:|:---:|
| 1 | 0.15 | 0.22 |
| 2 | 0.18 | 0.22 |
| 3 | 0.16 | 0.22 |
| 4 | 0.17 | 0.38 |
| 5 | 0.20 | 0.38 |
| 6 | 0.25 | 0.60 |
| 7 | 0.23 | 0.60 |

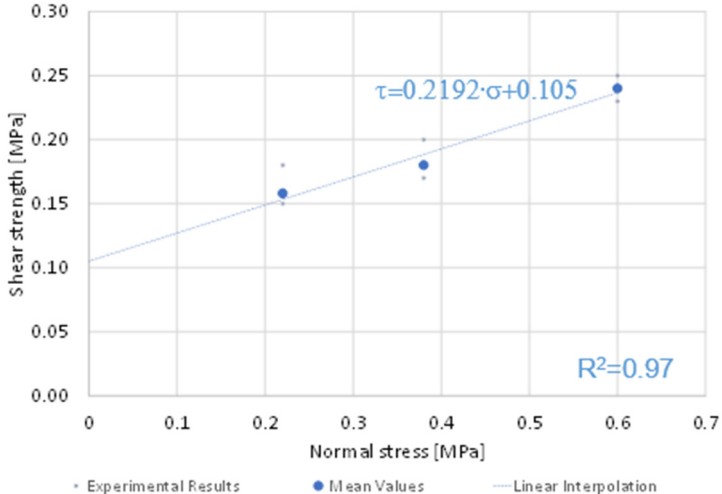

**Figure 9.** Direct shear test results. Results representation in the σ-τ stress plane.

## 3. The Numerical Model

### 3.1. General

Masonry buildings are complex types of constructions, made of inhomogeneous materials with a micro-structure characterized by units and joints. Different elastic and inelastic properties for each component of masonry walls make the behavior of the whole assemblage complex, with different failure mechanisms ruled by several parameters.

Several Finite element model (FEM) strategies were proposed to deal with numerical analyses of such complex material [30,31].

Among these, the modeling approach used in this paper is the continuous micro-model proposed in Petracca et al. [32]. This model is based on a tension–compression continuum damage model to accurately reproduce the nonlinear response of masonry elements, especially in shear. Units and mortar joints are modeled using 2D plane–stress continuum elements or 3D solid elements with nonlinear behavior, without resorting to interface elements.

### 3.2. Calibration of the Model-Based on the Direct Shear Tests on Triplet

The micro-modeling strategy described above has been used, for calibration purposes, to reproduce the direct shear tests treated in Section 2.4. The geometry of the triplet model is shown in Figure 10. Apart from the masonry triplet, it includes the parts of the supporting base in contact with the external stones, modeled as a rigid body, as well as the plate on the top of the central stone through which vertical displacement has been imposed. As real specimens, the masonry elements are composed of three units (limestone stone) and

two mortar joints. The dimensions of the units are 350 × 158 × 191 mm and all the mortar joints have a thickness of 10 mm.

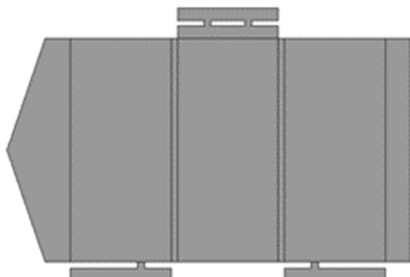

**Figure 10.** The Geometry of the Numerical model.

The model was described in the form of 2D plane stress elements available in GiD (v. 13.1.11 d). In fact, the analyses were performed using an enhanced version of the software Kratos Multiphysics, while pre- and post-processing were performed with GiD (v. 13.1.11 d), the graphical interface of Kratos used to model and to analyze the results. In order to save the computational cost of simulation, a regular mesh pattern whit a constant mesh size of 10 mm was used. The mechanical properties for the components are the same reported in Table 3 and are consistent with the mechanical features in Section 2.2.

**Table 3.** Material properties for units and mortar joints.

| *Materials* | $\gamma$ [kN/m$^3$] | $E$ [MPa] | $\nu$ [-] | $\sigma_t$ [MPa] | $G_t$ [N/mm] | $\sigma_0$ [MPa] | $G_c$ [N/mm] |
|---|---|---|---|---|---|---|---|
| Units | 25.5 | 20,000 | 0.20 | 0.80 | 0.01 | 80 | 9 |
| Mortar joints | 16.5 | 25 | 0.25 | 0.05 | 0.001 | 0.3 | 30 |

The pre-compression load was applied at the two lateral edges (steel plates) and kept constant during the analysis.

Finally, the vertical displacement was applied at the central unit top-up to masonry failure. The numerical analysis followed a full Newton–Raphson iterative procedure and convergence was accepted with a relative tolerance of the residual norm of $1.0 \times 10^{-5}$.

In Figure 11, the failure modes of the numerical analyses are shown. As it is possible to observe, they perfectly reproduce the sliding failures observed during the tests (see Figure 8).

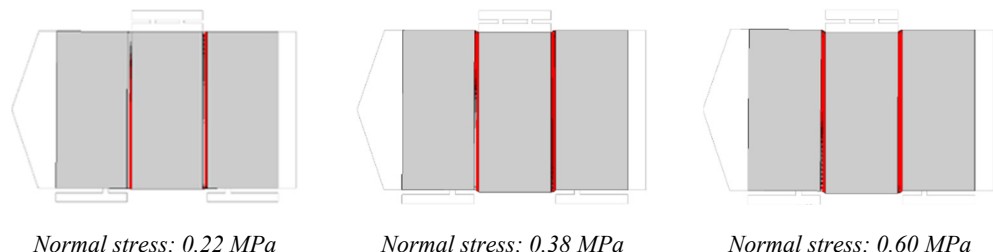

*Normal stress: 0.22 MPa*      *Normal stress: 0.38 MPa*      *Normal stress: 0.60 MPa*

**Figure 11.** The failure mode observed downstream of the numerical analysis.

The good matching between the numerical and the experimental models is also proven in Figure 12, where the maximum shear stresses are reported. From the analysis of the plotted results, it is also evident that the model is able to well follows the same Mohr–Coulomb failure criterium found via experimental analysis.

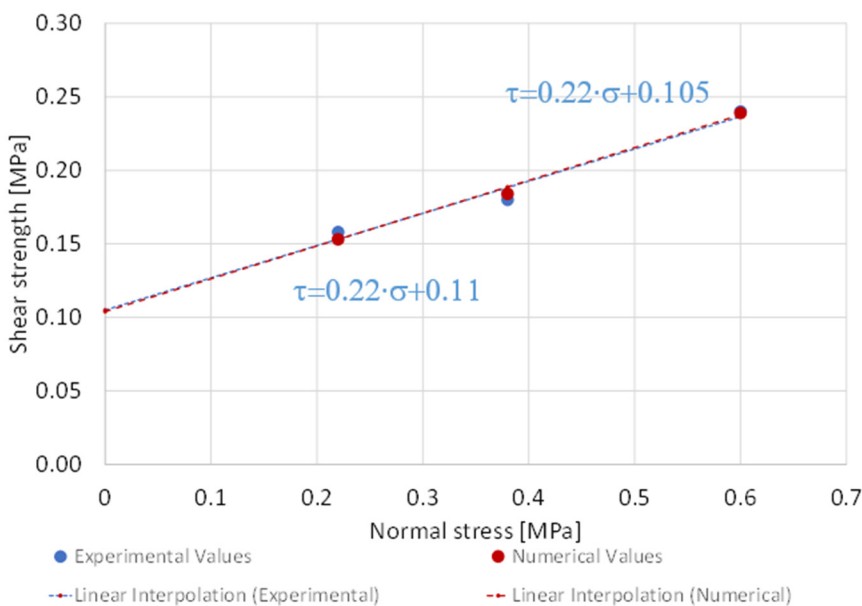

**Figure 12.** Direct Shear Tests: Comparison between numerical and numerical results.

## 4. Numerical Analyses of Unreinforced and Reinforced Masonry Panels in Shear
### 4.1. Basis

The good agreement between the numerical and the experimental results, shown in Section 3.2, allows us to consider the adopted modeling approach reliable enough to carry out other types of analyses for panels for which sliding along the mortar joint represents the most influencing failure phenomenon. Based on this premise, the response of panels in shear, made of the here studied masonry type, has been analyzed considering both the absence and the presence of a reinforcement system. The outcomes of these analyses are reported in the following paragraphs.

### 4.2. Numerical Analysis of Unreinforced Masonry Panels in Shear

Numerical analyses were performed on a $700 \times 700 \times 111.4$ mm masonry panel, adopting the same materials, the same sizes of the units and of the mortar joints used for simulating the direct shear tests. Furthermore, in this case, the software Kratos Multiphysics was used. In the pre-processing phase, the numerical model was constructed with 3D solid elements in a hexahedral configuration with 8-node, with three degrees of freedom (corresponding to the three translations) each.

The boundary conditions (BC) and the loading sequence were those reproducing the same condition prescribed by the Standard ASTM [33] for diagonal shear tests, which has been simulated as it represents the most diffused type of test in literature. In particular, horizontal and vertical displacements at the bottom vertex of the panel were totally restrained, while at the top part just the horizontal displacement was restrained. Then, the vertical displacement was applied at the panel top, up to masonry failure, with a velocity of 0.01 mm/s. The numerical analysis followed a full Newton–Raphson iterative procedure and convergence was accepted with a relative tolerance of the residual norm of $1.0 \times 10^{-5}$.

The result showed that no cracks involved the stones in the model, confirming one of main assumptions at the base of the analysis. In Figure 13, the shear stress–diagonal strain curve is shown, together with the cracked configurations of the panel at different shear demands. The maximum shear stress is about 0.074 MPa and was attained for a diagonal strain of 0.032, whereas the ultimate strain, considered when the strength was reduced by 20% of the maximum value, was attained for a vertical strain of 0.05.

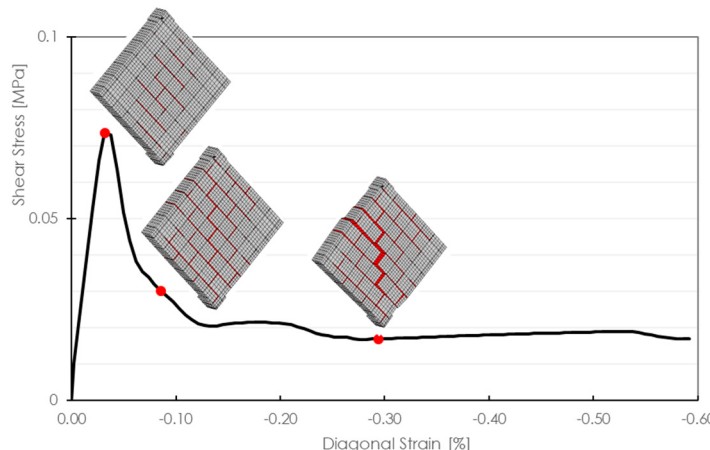

**Figure 13.** Shear stress–diagonal strain for the analyzed panel under diagonal compression.

Furthermore, in Figure 14, the principal stresses read for different shear demands are shown. It must be pinpointed that the stress concentration at the vertex of the panels is due to the fact that the model includes the model of the steel plates that are commonly used in laboratory tests for transferring the diagonal force.

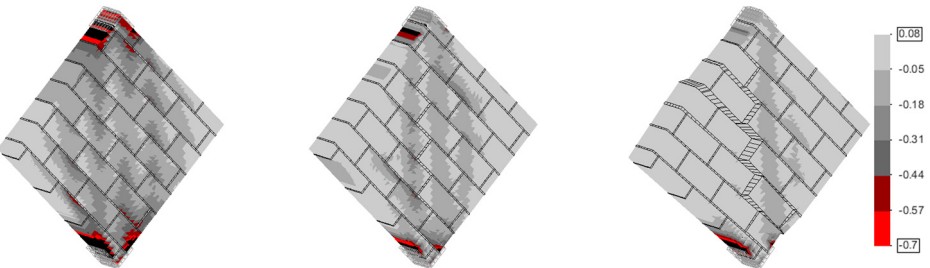

**Figure 14.** Evolution of principal stress in the masonry panel.

### 4.3. Numerical Analysis Reinforced Masonry Panels with FRCM System

The numerical model proposed in the previous Section proved that the studied masonry type leads to low shear strength masonry panels characterized by a brittle behavior. When a masonry wall presents this type of response, several retrofitting solutions are available. Some authors worked directly on the improvement of the mechanical features of traditional air lime mortar, by incorporating, in the admixture, fibers as reinforcement [34].

For example, Stefanidou et al. [35] added Posidonia Oceanica marine plant fibers, subjected to alkaline and hydrothermal treatment, to reduce the hygroscopicity and to improve the durability, with a final mix able to increase the flexural and compression strengths of about 55%. Izaguirre et al. [36] proposed the use of polypropylene fibers for reinforcing aerial lime mortar, finding a general improvement in terms of permeability, mechanical strengths, reduction in macroscopic cracks or durability in the face of freezing–thawing cycles. Seker et al. [37] used chopped carbon and glass fibers, obtaining increments of compressive strength up to 8% and up to 27% for the flexural strength.

On the other hand, other research studies were devoted to finding retrofitting techniques addressed to improve the structural response of the whole masonry panels in shear. Among these, one of the most effective is the one based on the application of composite materials, such as FRP (Fiber Reinforced Polymers), SRG (Steel Reinforced Grouts) and FRCM (Fiber Reinforced Cementitious Matrix) [38,39].

FRPs with organic matrices were the first to be widely used in the field of consolidation and rehabilitation of civil structures, due to the high tensile strength, the high strength/weight ratio, the easiness of application, the versatility, the affordable costs and the good corrosion resistance.

Furthermore, through the application of FRP bands on the external and/or internal faces of masonry walls, it is possible to achieve a considerable increase in shear strength because of the establishment of strut and tie mechanisms. However, despite the high performance, the main drawbacks encountered with the use of FRP are related to the poor compatibility with existing structures and their high sensitivity to impact and notches. In particular, in the case of the application on masonry substrates, the use of epoxy resins generates problems of mechanical compatibility and the transpiration of the masonry could be negatively influenced. The problem of mechanical compatibility is due to the significant differences in stiffness and strength of the substrate and of the reinforcement system, while the lack of breathability is due to the fact that epoxy resins produce a waterproof jacket that interrupts the flow of moisture in the masonry.

Both the critical issues discussed above could lead to premature degradation of the substrates: for this reason, the FRCM (Fiber Reinforced Cementitious Matrix) techniques, based on the use of inorganic matrices instead of epoxy resins, have been developed as a more sustainable solution. Inorganic matrices are particularly compatible with masonry in terms of adhesion with the substrate and they improve breathability, drastically reducing the issues related to mortar moisture. The FRCM reinforcement consists of the realization of a double layer of mortar that includes a bidirectional net realized with continuous fibers. It presents numerous advantages compared with traditional interventions, as FRCM reinforcement systems guarantee not only excellent chemical-physical and mechanical compatibility with masonry substrates, but also a low variation in terms of mass and stiffness of the structural element. For this reason, FRCM-based reinforcement have been selected as a promising intervention for the studied masonry elements and applied to the panel in shear studied in Section 4.2.

To this purpose, the model has been enriched by adding, on the two faces of the bare panel (Figure 15a), a first layer of mortar characterized by a thickness of 10 mm (Figure 15b) and having the mechanical features reported in Table 4. Then, the net, an AR-glass fibers net whose mechanical features are reported in Table 5, has been modeled through a continuous layer characterized by an equivalent thickness of 0.1 mm as described in CNR-DT 215/2018 (Figure 15c). Finally, another layer of mortar of 10 mm has been added on each face of the panel (Figure 15d). The final meshed model is shown in Figure 15e. All the layers have been tied rigidly, by imposing an internal constraint between the nodes on the external faces of the masonry and those ones of the juxtaposed layers, all characterized by the same mesh sizes.

**Table 4.** Material properties for mortar of the strengthening system.

| | $E$ [MPa] | $\sigma_t$ [MPa] | $G_t$ [N/mm] | $\sigma_0$ [MPa] | $G_c$ [N/mm] | $\varepsilon_p$ [-] |
|---|---|---|---|---|---|---|
| Mortar layer (NHL) | 12,000 | 0.9 | 0.7 | 8.0 | 50 | 0.05 |

**Table 5.** Material properties for AR-glass fibers net.

| | $E$ [GPa] | $\sigma_t$ [MPa] | $G$ [g/m$^2$] (Weight) | $g$ [g/cm$^3$] (Net Density) | $t_{eq}$ [mm] | $\varepsilon$ [%] |
|---|---|---|---|---|---|---|
| AR-glass fibers net | 67 | 1200 | 500 | 2.50 | 0.1 | 1.5 |

It must be pointed out that both the mechanical features of the mortar and of the AR-glass net have been taken by a study carried out by Miceli et al. [40], dealing with experimental tests of reinforced and unreinforced masonry panels in shear.

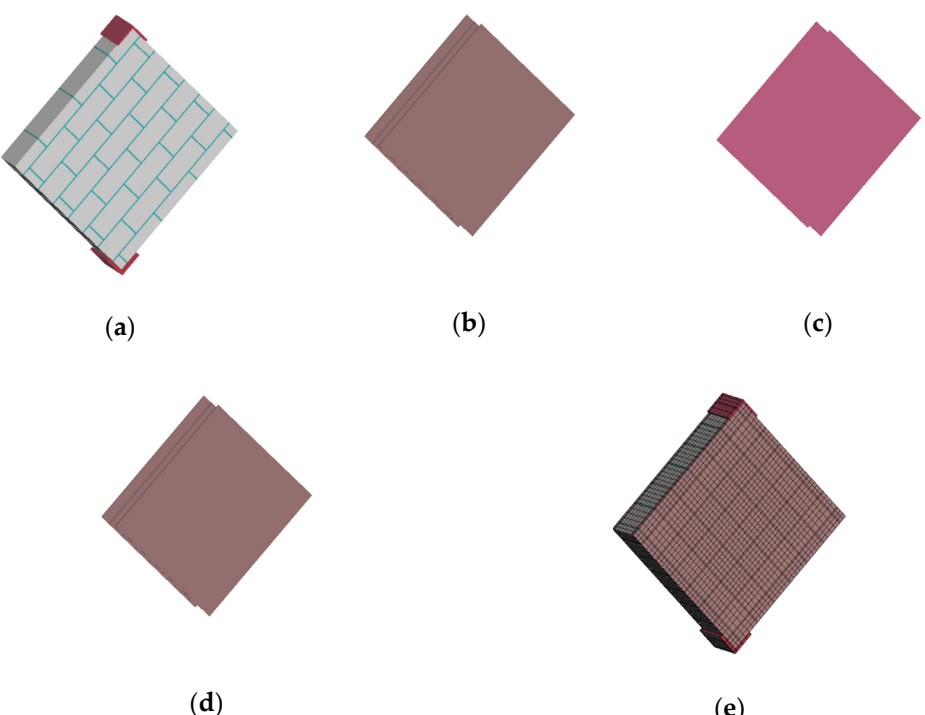

**Figure 15.** Construction of the numerical model of the FRCM reinforced shear panel: (**a**) unreinforced model; (**b**) first layer of mortar, ~10 mm; (**c**) equivalent thickness of AR-glass fibers net, 0.1 mm; (**d**) second layer of mortar, ~10 mm; (**e**) the meshed reinforced model.

Figure 16 shows the obtained results, under the same type of analysis carried out for the unreinforced panel. The FRCM-based reinforcement allowed us to obtain a maximum shear stress of 0.55 MPa and an ultimate shear stress, measured when the strength of the panels was reduced by 20% of 0.44 MPa, attained at a diagonal strain of 0.55. In the same figure, the evolution of the damage is depicted as well. As it is possible to observe, contrarily to the unreinforced solution, the sliding phenomena were reduced, and the failure was attained because of the crushing along the diagonal strut.

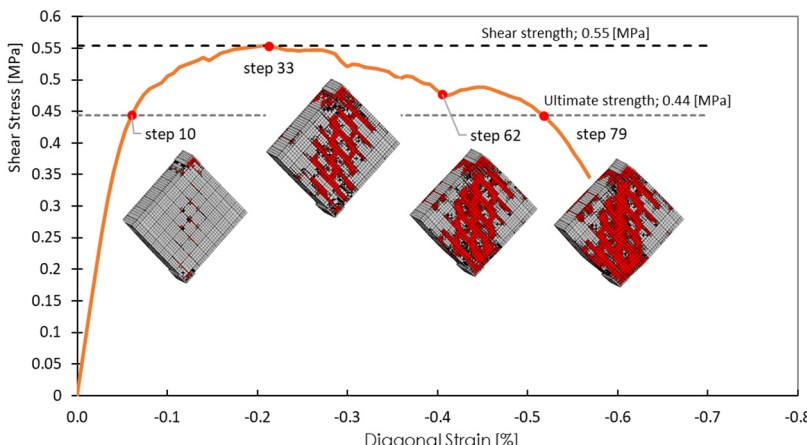

**Figure 16.** Shear stress–diagonal strain relationship (Model with complete FRCM reinforcement) and evolution of damage.

In Figure 17, the stresses of the reinforced panel are reported. In Figure 18, the comparison between the shear stress–vertical strain curves of the reinforced and unreinforced masonry panels in shear is depicted. For the sake of comparison, the curve provided by the

panels reinforced only with the two layers of mortar is reported as well, so to appreciate the contribution of the fiber reinforcement on the improved structural response.

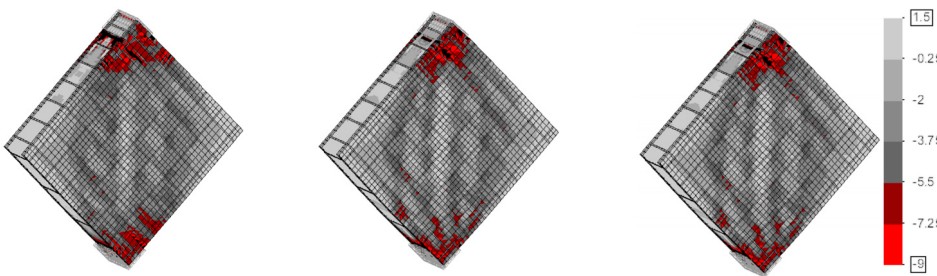

**Figure 17.** Evolution of principal stress in the FRCM reinforced masonry panel.

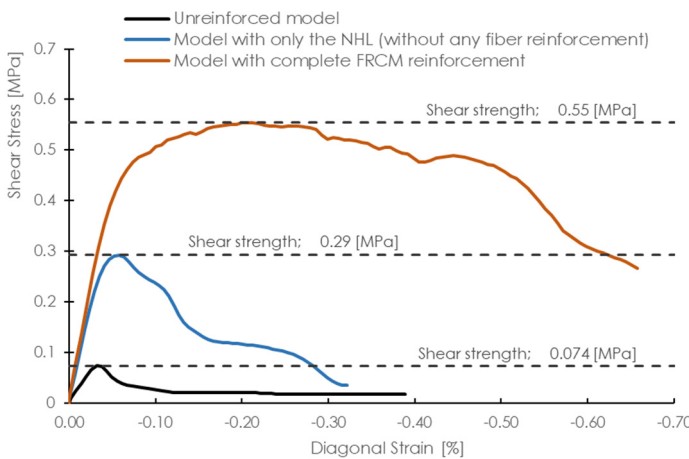

**Figure 18.** Shear stress–diagonal strain relationship (Model with complete FRCM reinforcement) and evolution of damage.

As it is possible to observe, the mortar jacketing leads to an increase of about four times both the peak and the ultimate strength, whereas the addition of the fiber reinforcement produces a further doubling (Figure 19).

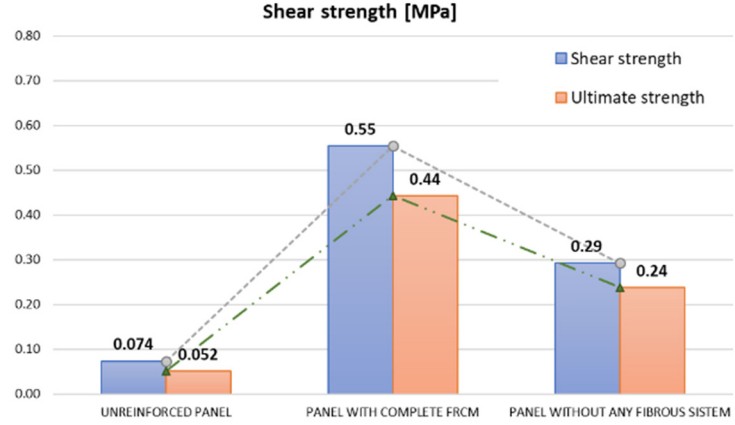

**Figure 19.** Shear stress–diagonal strain relationship (Model with complete FRCM reinforcement) and evolution of damage.

However, the most important contribution of the fiber reinforcement is in terms of ductility, with an ultimate diagonal strain that is increased by almost three/four times for the panel with the mortar layer only and thirteen times if the fiber net is considered further.

## 5. Conclusions

This paper dealt with a low-strength mortar masonry, which, although manufactured in a laboratory, presents the same mechanical features that can be usually found in old buildings belonging to ancient historical centers.

Experimental tests, repeated for different curing times and carried out to mechanically characterize both the stone units and the mortar, have been presented. Moreover, direct shear tests on triplets have been executed, under different axial forces, in order to analyze the shear sliding response of the studied masonry.

On the basis of direct shear tests, a numerical model of the studied masonry has been implemented and used to evaluate the shear response of a masonry panel both in the unreinforced and in the reinforced configuration. The obtained results proved that the considered FRCM-based reinforcement, apart from representing a sustainable solution characterized by a high compatibility with the masonry support, is really effective in increasing both the strength and the ductility.

The main findings of the paper can be summarized as follows:

- The proposed mortar mix, reported in Table 1 and studied in detail through XRD and TGA analyses, allows a compression strength of about 0.30 MPa, which is truly a low value, when compared with the low strength of some mixtures proposed in several studies in the literature. This value is really comparable with the strengths characterized the degraded mortar of old buildings;
- The proposed numerical modeling approach, based on a tension–compression continuum damage model, is able to well-represent the shear behavior of the proposed masonry, provided that this is ruled by the poor shear strength of the mortar;
- The studied masonry type can be profitably reinforced through FRCM-based retrofitting interventions, which leads to an increment of shear strength of about eight times the strength of the unreinforced masonry;
- Most importantly, FRCM produces an increment of ductility thirteen times with respect to the ductility of the unreinforced masonry.

**Author Contributions:** Conceptualization, G.B. and E.S.; methodology, G.B.; software, G.V. and F.D.M.; validation, G.B. and E.S.; formal analysis, G.V. and F.D.M.; investigation, G.B., I.C., G.V. and F.D.M.; resources, G.B. and E.S.; data curation, I.C., G.V. and F.D.M.; writing—original draft preparation, F.D.M.; writing—review and editing, G.B.; visualization, E.S.; supervision, G.B. and E.S.; project administration, G.B. and E.S.; funding acquisition, G.B. and E.S. All authors have read and agreed to the published version of the manuscript.

**Funding:** This research was funded by the Italian Research Project ReLuis 2018–2021, funded by the Italian Department of Civil Protection.

**Institutional Review Board Statement:** Not applicable.

**Informed Consent Statement:** Not applicable.

**Data Availability Statement:** The authors state their availability in sharing their results, stored at the University of Chieti-Pescara, under request.

**Acknowledgments:** Tests were carried out at the SCAM laboratory of Chieti-Pescara. Tests have been carried out through DeWesoft Instruments. The authors thank Fabio Iucolano of the Department of Chemical, Materials and Industrial Production Engineering of University of Naples Federico II, for carrying out XRD and TGA analyses.

**Conflicts of Interest:** The authors declare no conflict of interest.

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
