# Peer review of "Experimental and Numerical Mechanical Characterization of Unreinforced and Reinforced Masonry Elements with Weak Air Lime Mortar Joints"

_sustainability, doi:10.3390/su14073990_

Round 1

Reviewer 1 Report

The following need to be done:

-Improve the introduction by adding the significance of your research

-Put more information of materials

-Figure 1 on page 5? is it correct? wrong numbering

-Figure 4 not scientific revise

-Figure 7 is very basic presentation

-Discussion and conclusion is very brief need extra information

-

Author Response

Authors1#0: We thank the Reviewer for her/his time in the review process. According to the points posed during the review, we improved the following aspects:

_______

Reviewer1#1: Improve the introduction by adding the significance of your research

Authors1#1: Thank you for the suggestion. We added at the end of Section 1 the following text: ““To the best of our knowledge, the paper deepens a type of masonry, with very weak mortar joints, that, although a certain literature on the topic exists, was investigated seldom, in particular in laboratory environment, where obtaining very low strength of the mortar (about 0,5 MPa) has represented often a challenging objective. Furthermore, the evaluation of the effectiveness of FRCM based retrofitting intervention on this type of masonry definitely represents an element of significance relevance in the considered research field, as well as an important advancement with respect to the current literature.”

_______

Reviewer1#2: Put more information of materials

Authors1#2: Actually, we carried out XRD and Thermal analyses in order to assess the mineralogical composition of the mortar and the thermal behavior. In the original version of the paper, these analyses were not reported because the study was more focused on the structural features of the masonry and of its components. On the other hand, we realized that their description could help the reader to better understand the mortar peculiarities, which, indeed, represent one of the novelties of the paper. We decided to add the text reported in red after Table 1.   

_______

Reviewer1#3: Figure 1 on page 5? is it correct? wrong numbering

Authors1#3: Sorry, there was a problem related to the transformation of the doc document in pdf. We corrected that and we checked again the pdf before the new submission

_______

Reviewer1#4: Figure 4 not scientific revise

Authors1#4: We homogenized figure 4 to figure 3, assuming that the latter was good for the Reviewer.

_______

Reviewer1#5: Figure 7 is very basic presentation

Authors1#5: Thank you. We revised that. Please, consider that we found an error in the transcription of some data. They have been slightly modified with respect to the original version of the paper.

_______

Reviewer1#6: Discussion and conclusion is very brief need extra information

Authors1#6: Thank you, we enriched the Section Conclusion, by giving more detailed information on the obtained results and stressing the novelties introduced by the paper. To this purpose, we bullets points were used to synthesize the achieved outcomes. As it was the request of another Reviewer, we left the text in black.

______

All the changes carried out according to Reviewer#1’s query can be traced in the paper colored in red

Reviewer 2 Report

First of all, the manuscript is quite well prepared, the topic is interesting and provides new knowledge. Nevertheless, there are some points that need to be addressed. Specifically, the key words are quite big almost phrases and these are difficult to be searched for as key words. In line 31, probably you had the intension to write literature "review" or "analysis"? In line 33, you rather combine the refereces in one brackets parenthesis such as [4-6] following the format of the journal, as well as applying that in the whole text. In my opinion, the description of state-of-the-art could be improved, using 3 or 5 more relevant articles. I would propose also to the authors to find significant information in DOI: 10.1016/j.conbuildmat.2021.123881 and incorporate them in the text in the theoretical approach. In line 99, please make the title more simple in order the reader to understand that the experimental part/materials and methods begins. It would be useful to provide also a figure of the prepared specimens. Provide the manufacturer/origin of superplasticizer used. In line 128, please clarify the "target low strength". In line 133, are there any differences in the italian standard compared to the same european standard? Since it is crucial to be easily repeated by the research community (in the case that there are no differences, please keep the EN name). In the caption of Figure 4, you rather keep only the phrase bending strength and remove the phrase tensile strength in order to be more accurate and clear to the reader (since the figure is not very distinct). Check again if it acceptable by the journal to combine the experimental part (materials-methods) with the results section. You should describe (optimally in materials-methods) also the statistical analysis method applied to the results. I could not find such a description.

Author Response

Reviewer2#0: First of all, the manuscript is quite well prepared, the topic is interesting and provides new knowledge. Nevertheless, there are some points that need to be addressed.

Authors2#0: We really thank the Reviewer for his/her gratifying comment. We considered all the posed queries and a general improvements of the paper was achieved. The replies to each point are reported in the following and highlighted in green in the new revised version of the paper:

_______

Reviewer2#1: Specifically, the key words are quite big almost phrases and these are difficult to be searched for as key words.

Authors2#1: Thank you, we reduced the number of keywords and we reduced the length of those remaining

_______

Reviewer2#2: In line 31, probably you had the intension to write literature "review" or "analysis"?

Authors2#2: As correctly required, we added “analysis” to the text

_______

Reviewer2#3: In line 33, you rather combine the references in one brackets parenthesis such as [4-6] following the format of the journal, as well as applying that in the whole text.

Authors2#3: Thank you. We changed the format.

_______

Reviewer2#4: In my opinion, the description of state-of-the-art could be improved, using 3 or 5 more relevant articles. I would propose also to the authors to find significant information in DOI: 10.1016/j.conbuildmat.2021.123881 and incorporate them in the text in the theoretical approach.

Authors2#4: We really thank the Reviewer for the suggested articles, which allowed to improve the state of the art concerning those interventions devoted to improve the mortar mechanical feature. In Section 4.3, we added the following text:

Some Authors worked directly on the improvement of the mechanical features of traditional air lime mortar, by incorporating, in the admixture, fibers as reinforcement [33]. For example, Stefanidou et al. [34] added Posidonia Oceanica marine plant fibers, subjected to alkaline and hydrothermal treatment, to reduce the hygroscopicity and to improve the durability, with a final mix able to increase the flexural and compression strengths of about 55%. Izaguirre et al. [35] proposed the use of polypropylene fibers for reinforcing aerial lime mortar, finding a general improvement in terms of permeability, mechanical strengths, reduction in macroscopic cracks or durability in the face of freezing-thawing cycles. Seker et al. [36] used chopped carbon and glass fibers, obtaining increments of compressive strength up to 8% and up to 27% for the flexural strength.

On the other hand, other researches were devoted to find retrofitting techniques addressed to improve the structural response of the whole masonry panels in shear. Among these….

_______

Reviewer2#5: In line 99, please make the title more simple in order the reader to understand that the experimental part/materials and methods begins..

Authors2#5: Thank you. We revised the title as following: “The experimental tests”. Actually, it is more than sufficient

_______

Reviewer2#6: It would be useful to provide also a figure of the prepared specimens.

Authors2#6: We added a figure (Figure 4), according to the Reviewer query.

_______

Reviewer2#7: Provide the manufacturer/origin of superplasticizer used.

Authors2#7: The manufacturer is now specified in Table 1.

_______

Reviewer2#8: In line 128, please clarify the "target low strength".

Authors2#8: Actually “target” is ambiguous. We replaced it with “desired”

_______

Reviewer2#9: In line 133, are there any differences in the italian standard compared to the same european standard? Since it is crucial to be easily repeated by the research community (in the case that there are no differences, please keep the EN name).

Authors2#9: Thank you. No differences between the Italian and the European Standards. We quoted the European one only.

_______

Reviewer2#10: In the caption of Figure 4, you rather keep only the phrase bending strength and remove the phrase tensile strength in order to be more accurate and clear to the reader (since the figure is not very distinct).

Authors2#10: The Reviewer is right. Thanks.

_______

Reviewer2#11: Check again if it acceptable by the journal to combine the experimental part (materials-methods) with the results section.

Authors2#11: We checked and we did not find anything about to combine/to not combine. We preferred to leave the general organization of the paper as it was. Thanks.

_______

Reviewer2#12: You should describe (optimally in materials-methods) also the statistical analysis method applied to the results. I could not find such a description.

Authors2#12: Thank you, the statistical method used for the sigma-tau law is now described in Section 2.4 (in blue, to reply to another Reviewer).

All the changes carried out according to Reviewer#1’s query can be traced in the paper colored in green

Reviewer 3 Report

The paper presents a good topic related to experimental and numerical mechanical characterization of un-2 reinforced and reinforced masonry elements with weak air lime 3 mortar joints. The attached file contains many comments to improve the paper.

Author Response

Reviewer3#0: The paper presents a good topic related to experimental and numerical mechanical characterization of un-reinforced and reinforced masonry elements with weak air lime mortar joints. The attached file contains many comments to improve the paper.

Authors3#0: We really thank the Reviewer for the time spent for analyzing the paper, as well as for the interest stated on the topic. We accepted all the posed suggestions/queries -as reported below in our replies- and a general improvement of the paper has been achieved consequently. In order to make easy the track of the required changes, we reported in blue the modified/added parts of the text.

_______

Reviewer3#1: Page 1. Abstract must be rewritten seriously, such as the new findings and novelty. The summary needs to contain more quantitative information.

Authors3#1: Thank you, we integrated the abstract by adding more quantitative information. As for the new findings, also considering the query of another Reviewer, we reported those at the end of the Section “1. Introduction” (text in red)

_______

Reviewer3#2: Page 1. The keywords should include the most important results

Authors3#2: Thank you. This was a query posed also from another Reviewer. We revised the keywords, excluding those one that were too general and making more meaningful the remaining ones

_______

Reviewer3#3: Page 2. The authors should provide better explanation for the readers what was the difference between existing literature and the present study.

Authors3#3: Thank you. As stated before, at the end of the Section “Introduction”, we introduced, also because of the comments of another Reviewer, the following text (in red):

To the best of our knowledge, the paper deepens a type of masonry, with very weak mortar joints, that, although a certain literature on the topic exists, was investigated seldom, in particular in laboratory environment, where obtaining very low strength of the mortar (about 0,5 MPa) has represented often a challenging objective. Furthermore, the evaluation of the effectiveness of FRCM based retrofitting intervention on this type of masonry definitely represents an element of significance relevance in the considered research field, as well as an important advancement with respect to the current literature.

_______

Reviewer3#4: Page 5. check figure number.

Authors3#4: Thank you we checked all the numbers, because we found some error probably due to the conversion from doc to pdf. Now, numbers of figures and tables should be correct.

_______

Reviewer3#5: Page 6. where the fig 5a , 5b.

Authors3#5: You are right. We amended that

_______

Reviewer3#6: Page 6. The authors should provide better explanation.

Authors3#6: Thank you. We improve Section 2.4 by adding more information on the tests and on the obtained results.

_______

Reviewer3#7: Page 13. Please check all Table numbers.

Authors3#7: See Authors3#4.

_______

Reviewer3#8: Page 13. Add some of the important findings to conclusion

Authors3#8: Thank you, as replied to another Reviewer who posed the same query, we enriched the Section Conclusion (we left the text in black), by giving more detailed information on the obtained results and stressing the novelties introduced by the paper. To this purpose, we bullets points were used to synthesize the achieved outcomes.

Round 2

Reviewer 1 Report

I highly recommend to revise the graphical presentation of results

Reviewer 2 Report

As I have checked the authors have implemented the proposed changes in the revised verion of manuscript towards the improvement of their work. Almost all the changes have been implemented and in my opinion, the manuscript is well-prepared and organized enough to be accepted for publication in this journal. I remain at your disposal for any clarification.